# Macro-B12 and Unexpectedly High Levels of Plasma B12: A Critical Review

**DOI:** 10.3390/nu16050648

**Published:** 2024-02-26

**Authors:** Sergey N. Fedosov, Ebba Nexo

**Affiliations:** 1Department of Molecular Biology and Genetics, Aarhus University, 8000 Aarhus C, Denmark; 2Department of Clinical Medicine/Biochemistry, Aarhus University Hospital, 8200 Aarhus N, Denmark

**Keywords:** cobalamin, haptocorrin, PEG precipitation, routine testing, transcobalamin

## Abstract

A low total plasma vitamin B12 supports a clinical suspicion of B12 deficiency, while the interpretation of an unexpectedly normal/high level is marred by controversies. Here, we critically review current knowledge on B12 in blood plasma, including the presence of the so-called “macro-B12”. The latter form is most often defined as the fraction of B12 that can be removed by precipitation with polyethylene glycol (PEG), a nonspecific procedure that also removes protein polymers and antibody-bound analytes. Plasma B12 includes B12 attached to transcobalamin and haptocorrin, and an increased concentration of one or both proteins almost always causes an elevation of B12. The total plasma B12 is measured by automated competitive binding assays, often incorrectly referred to as immunoassays, since the binding protein is intrinsic factor and not an antibody. An unexpectedly high level of B12 may be further explored using immunological measurements of haptocorrin and transcobalamin (optionally combined with e.g., size-exclusion chromatography). Nonspecific methods, such as PEG precipitation, are likely to give misleading results and cannot be recommended. Currently, the need for evaluation of a high B12 of unknown etiology is limited since other tests (such as measurements of methylmalonic acid) may better guide the diagnosis of B12 deficiency.

## 1. Introduction

Vitamin B12 (cobalamin, B12) deficiency may result in irreversible neurological damage. Therefore, an early diagnosis is mandatory, most often guided by the measurement of total B12 in a blood sample [1,2].

Measurements of other biomarkers, such as the “active” part of B12 (holoTC or “active B12”), or the metabolic marker methylmalonic acid (MMA) and, to some extent, homocysteine (Hcy), are considered superior to total B12 in their diagnostic utility [2,3,4]. Yet, the measurement of total plasma B12 remains the first-line analysis when suspecting B12 deficiency. This is due to the easy availability of the assay and its low cost, as compared to the assessment of other analytes.

A low level of plasma total B12 suggests the presence of a deficient state, while the interpretation of a higher level may be a challenge. Does an unexpectedly high level reflect the absence of B12 deficiency, or should the clinician be aware of other possibilities?

Numerous case studies, as well as more extensive works on cohorts of patients, have addressed this issue, giving somewhat conflicting conclusions. Some authors argue that a high level of total plasma B12 (observed without any clear reason) has limited clinical implications [5]. Others suggest a high level of B12 is misleading and is caused by analytical flaws, or the presence of an ill-defined inert “macro-B12” that can be removed by precipitation with polyethylene glycol (PEG) [6,7,8]. Some papers go so far as to suggest an algorithm involving PEG precipitation applied to all samples with an increased level of B12 [9,10,11]. Much of these controversies are caused by an incomplete understanding of the nature of B12 and its binding proteins in blood plasma, as well as the analytical procedures employed for its analysis.

Here we outline the current knowledge on B12 and its binding proteins in plasma. We also describe methods used for measurement of the vitamin and for identifying the causes of a surprisingly high level of B12. Finally, we provide our recommendations for handling any unexpected occurrences of an elevated B12 in plasma.

## 2. Plasma B12 and Its Binding Proteins

Plasma B12 (total B12) covers various forms of the vitamin bound to two binding proteins, haptocorrin (HC) and transcobalamin (TC) (Figure 1) [12]. In healthy individuals, the concentration of total B12 is around 200–600 pmol/L [13]. The basic features of B12 and the interplay between its transporters are briefly outlined below.

The structure of B12 is presented schematically in Figure 1 (red sketch in the middle). The upper ligand “X” (bound to the Co-ion) differs in various molecular forms of the vitamin: the coenzymes methyl-B12 and deoxyadenosyl-B12, as well as their ubiquitous precursors, e.g., hydroxo-B12 and cyano-B12, are abundant in foods and vitamin pills, respectively. All four forms are present in blood plasma and bind with comparable affinities to the B12 binding proteins [14,15]. In addition, plasma contains inactive B12 derivatives (the so-called analogs [16]) without a known biological function in humans.

Approximately 4/5 of circulating B12, as well as all of the analogs, are bound to HC. In healthy individuals, HC occurs at a concentration of 240–680 pmol/L [17]. HC is a glycoprotein produced in all exocrine glands and in white blood cells. Its half-life (approximately 17 days) is quite long ([18] and refs. thereof). The turnover of HC becomes considerably faster if the content of sialic acid on its surface is low, as is the case with HC encapsulated in granulocytes [19]. B12 and its analogs bound to HC are slowly cleared by the liver, where HC is degraded, and the attached ligands are excreted with the bile. The main part of the excreted B12 is reabsorbed after binding to intrinsic factor (IF), a protein produced by the parietal cells in the stomach [1]. IF recognizes B12 and facilitates its intestinal uptake via interaction with an IF-B12-specific receptor (Figure 1). At the same time, IF has a low affinity for the analogs, which are expected to pass to the large intestine without any significant adsorption [18].

B12 from dietary sources eventually enters the blood, peaking at approximately 7 h after ingestion (see Figure 1 and ref. [18] for a review). The incoming vitamin primarily binds to TC due to its high binding capacity (600–1200 pmol/L [18,20]), maintained by a steady production of TC in endothelial cells. TC has an affinity for B12, which is comparable to that of HC, but the binding of the analogs to TC is relatively weak. The produced TC-B12 complex (often referred to as holoTC or “active B12”) is promptly absorbed by human tissues (t_½_ ≈ 1 h [21]) and particularly by the liver. The fast clearance of TC-B12 stipulates its low steady-state concentration (≈60–80 pM), which corresponds under normal conditions to 10–20% of total plasma B12. A low level of holoTC contrasts with the high concentration of the “inert” HC-B12 complex, and this disproportion occasionally creates confusion during the assessment of B12 status by measuring total plasma B12. Therefore, a separate test for the amount of holoTC in blood is currently implemented in many medical laboratories [1,2,3,4].

Total B12 is the sum of B12 bound to TC and HC in both healthy individuals and in individuals with an unexpected level of total B12 [22]. There is one exception, however. Excessive hydroxo-B12 may be retained in the body through a weak complex formation with other proteins, e.g., albumin [23], which in turn explains the very high level of plasma B12 in patients receiving injections of this form of the vitamin. Any free B12 that exceeds the specific or unspecific binding capacity of the blood proteins is promptly excreted through urine [24,25]. Therefore, an elevated level of B12 frequently originates from an increased concentration of circulating HC or TC, or both. These two very strong binders accumulate all B12 if their total concentration exceeds that of B12.

A high level of HC and/or TC may be caused by an increased synthesis of these proteins, as exemplified by patients with cancer cells producing HC [22] or TC [26]. Another reason for a high level of TC is the presence of substances that decrease the turnover rate of the protein, such as autoantibodies [27].

## 3. Measurement of Plasma B12

Total plasma B12 was initially measured by either microbiological assays or in-house competitive isotope dilution assays, some of which recognized not only B12 but also its analogs [16,28]. Today’s methods are competitive binding procedures specific for B12 and suitable for automatic platforms.

The automated determination of B12 is often referred to as a competitive immunoassay. Yet this classification is misleading because the key B12-capturing compound is not an antibody but the B12 binding protein IF that binds B12 (but not its analogs) with an affinity of around 10^14^ L/mol [18].

Before reaching the competitive part of the assay, all B12 is released from its binding proteins and converted to the cyano-form of B12. This is achieved by exposure of the sample to alkaline pH combined with the reducing agent DTT and cyanide ions. It should be noted, that if any B12 binding protein is resistant to this treatment or renatures upon the obligatory neutralization prior to B12 measurement, obviously, it might have the potential to interfere with the assay.

The methodological details differ according to the analytical platforms [29]. Table 1 summarizes the design for the commonly employed platforms of Siemens, Roche, Abbot, and Beckman Coulter. Only Access from Beckman Coulter uses an antibody, and only as a secondary reagent.

The ranges of measurement and the reference intervals are not identical for different assays. It should also be noted that the values above the upper measurement limit usually are presented as e.g., >1476 pmol/L, and the samples are diluted only upon request by the clinician. Therefore, very high values of B12 might remain undisclosed.

## 4. “Macro-B12” as an Explanation for an Unexpectedly High Plasma Total B12

Several studies question whether a normal or a high level of total B12 is a correct reflection of the true B12 status or indicates the presence of an inert “macro-B12”, without giving any strict definition of the latter expression.

The general term “macro-protein” covers any polymer of the analyte [30] or a complex between the analyte and other proteins, notably antibodies [7,31,32]. The presence of such macro-forms is most often of no impact on the biological function of the analyte but may result in spurious—most often—high values.

B12 would be present in a “macro” form in those cases, where either TC or HC, or B12 itself, binds to a substance of a high molecular weight. Such complex formation is in general unlikely to have a negative consequence for the normal physiology of B12 uptake by the tissues. For example, “macro-TC-B12” is not a dead end for B12 turnover because this form exists in a reversible equilibrium with free TC-B12 [33] according to the below scheme:(plasma macro TC-B12) ↔ (plasma TC-B12) ↔ (tissue B12)

In other words, “macro B12” does not necessarily mask a B12 deficiency, being a constituent of a B12 exchanging pool. A constant influx–efflux of TC-B12 would eventually bring the system to a steady state, where the level of free TC-B12 in plasma is buffered from both sides of the equilibrium. In addition, the quantity of B12 in the tissues (total body B12 ≈ 1.5–3.7 μmol = 2.0–5.0 mg) is at least 1000-fold higher than the normal level of B12 in plasma (1–1.5 nmol), see e.g., Table 6-15 in ref. [34]. This situation does not significantly change even if considering a 10-fold excess of macro TC-B12 in plasma. Consequently, the buffering capacity of the tissues conferred on free plasma TC-B12 exceeds by far that of macro TC-B12. Nevertheless, symptoms of B12 deficiency have been described to occur in a few patients, where macro-B12 was judged to hide a B12 deficiency, but with insufficient data to judge the nature of the macro-B12 [35,36].

Observations of well-documented cases of macro-proteins carrying plasma B12 are relatively rare. Most of them represent case studies demonstrating macro-TC, i.e., TC that interacts with antibodies having some affinity for TC [33,37]. Such antibodies may be induced by B12 injection therapy [33,38] but have also been reported as an unexplained finding in untreated individuals [6]. Notably, no study reports a biological effect related to the presence of macro-TC. The general understanding is that the only effect of such complex formation is a decrease in the clearance rate of TC-B12, not influencing the delivery of B12 to the cells.

More recently, the soluble receptor for TC (sCD320) has been shown to form complexes with TC in plasma [39], and a family with high levels of B12, TC, and CD320 has been described, yet without any impact on the metabolism of B12 [40].

Only a few cases of macro-HC have been described [41,42], and likewise, only a single case of gamma globulin directly recognizing free B12 has been documented [43]. None of these publications presented any impact on the B12 supply to the cells of the body.

Unfortunately, the term macro-B12 has not been limited to well-documented cases demonstrating the presence of macro-TC, macro-HC, or possibly complexes between other proteins and B12. Mainly in recent years, the term has frequently been used to indicate substances removed by a nonspecific precipitation methodology, such as polyethylene glycol (PEG). A decreased level of B12 following PEG precipitation has been reported in up to 45% of samples with initial B12 levels above 554 pmol/L, and this in turn led to the suggestion that such samples should undergo PEG precipitation prior to reporting a result for B12 [9]. However, the substances removed by PEG precipitation are unknown, as is the possible impact of such substances on B12 metabolism; see the next section for further discussion.

## 5. Methods Used to Identify Macro-B12

### 5.1. PEG Precipitation

Polyethylene glycol (PEG) is widely used to remove unwanted antibodies and macromolecules from plasma samples. The method depends on the elimination of fluid from the sample, thereby inducing the less soluble proteins to precipitate. In other words, this is a nonspecific methodology that should be used with caution [44,45].

The fraction of macro-TC-B12 in the equilibrium (antibody macro-TC-B12) ↔ (free TC-B12) significantly varies depending on pH, ionic strength, and temperature [33]. Consequently, the percentage of PEG-precipitated B12 might differ depending on the chosen conditions. In addition, TC is prone to precipitation, particularly while in complex with B12-forms other than hydroxo-B12 [46], and it is partly removed by PEG in healthy individuals [47]. Thus, PEG precipitation may well remove a fraction of B12 fully available for the cells. Based on the aforementioned reasons, we find advocating for the use of PEG precipitation prior to B12 measurements to be misleading and not advisable.

### 5.2. Silica Gel Precipitation

Silica gel has been employed because of its ability to precipitate TC while HC remains in the supernatant [48]. Silica gel precipitation is a nonspecific procedure that separates molecules based on their charge. The pI of TC is nearly neutral 6.29–6.75 [49,50], while the glycoprotein HC has an expressively acidic pI of approximately 3.1–3.3 [50,51]. Nonspecific precipitation with silica gel may cause a misinterpretation of the obtained result, as was the case in a recent study that wrongly explained a high level of B12 by its retention on HC, while it was caused by the presence of macro-TC, not precipitated with silica gel [52].

### 5.3. Ammonium Sulfate Precipitation

Ammonium sulfate is widely used for antibody precipitation [52,53], although the procedure also precipitates 14% of TC [51], and the results obtained with this method should therefore be interpreted with reservations.

### 5.4. Size Exclusion Chromatography

Early studies relied on size exclusion chromatography to separate TC and HC, as well as to document the presence of macro-B12 that elutes earlier than both proteins [54]. At the time, it was noted that the recovery of TC in its free form (and not as a high molecular compound) required well-defined conditions, such as a temperature of 20–37 °C and a buffer with a high ionic strength and/or pH 4.5 [33,54,55]. In more recent studies, the buffers used are poorly described, and because of this, it is difficult to judge the accuracy of the data [7,56].

### 5.5. Gamma Globulin Specific Agents

A few studies depend on the use of gamma globulin-specific agents, such as protein G, to show macro-B12 containing gamma-globulins [8,43]. Such methods are judged to be more specific than other precipitation methods. However, it remains to be shown whether the rather “sticky” TC binds to any such agents before a firm conclusion about their usefulness can be made.

### 5.6. ELISA Methods

ELISA methods, capable of quantifying TC and HC, are now available. These methods are preferable if a clarification of an increased level of B12 is requested [40]. A combination of immunological assays with size exclusion chromatography may provide a further specification of the protein species carrying B12.

### 5.7. Conclusions on the Occurrence of Macro-B12

Most studies on the presence of macro-B12 performed so far employ nonspecific methodologies and do not allow for any conclusion as to the frequency of macro-B12 or its importance in the clinical setting.

## 6. Other Possible Explanations for an Unexpectedly High Level of Plasma B12

Obviously, the most important issue, when judging equivocal results for plasma B12, is to rule out analytical errors resulting in a falsely elevated level of B12.

### 6.1. Autoantibodies against Intrinsic Factor

Autoantibodies against intrinsic factor are detected in some patients with autoimmune gastric atrophy (pernicious anemia) [1]. Notably, antibodies that block binding of B12 to IF would be able to cause a falsely increased level of B12 if they survive the harsh pretreatment of the samples prior to measurement of B12.

An interference by autoantibodies was inferred in studies showing that B12 declines after precipitation with PEG [57] or with protein G [8,43]. In contrast, a study systematically examining the influence of IF autoantibodies on several B12 assays from Siemens, Roche, and Beckman Coulter reported no interference [58].

At present, we cannot rule out that autoantibodies against IF may cause deviating results. Yet, studies using stringent methodologies to document such an effect, are lacking.

### 6.2. Heterophilic Antibodies

Heterophilic antibodies are relatively common and may give spurious results because they react with animal gamma globulins used in automated immune assays. Yet, several publications document no interference from such antibodies in B12 assays [6,7,10]. This is not surprising because B12 measurements depend on IF and do not involve antibodies during B12 capturing. Therefore, testing for heterophilic antibodies is meaningless for B12 tests, possibly except for the Access assay that uses a goat anti-mouse anti-IF antibody to bind labeled IF that remains unsaturated after its interaction with B12 from the sample (see Table 1).

### 6.3. Extreme Concentrations of Haptocorrin

Very high HC may interfere in assays of B12 because a part of HC remains intact even after the harsh pretreatment procedures. This in turn may result in a falsely high value of B12 (the Siemens test) or falsely low level (the Abbott test). This interference has been documented for HC levels of 10 nmol/L and above [59], which is high compared to the reference values for HC of 240–680 pmol/L [17]. Such a high concentration is present in human milk [59], but seldom in plasma. In addition, extreme values of HC may result in a high level of B12 despite a deficient state because the excessive HC hinders the binding of B12 to TC and thereby decreases the vitamin available for the cells.

## 7. Recommendations for Handling of Unexpectedly Normal/High Levels of Plasma Total B12

The clinician is challenged when a normal/high level of plasma B12 occurs in a patient with symptoms suggesting B12 deficiency. The highest priority is obviously to clarify whether the symptoms are caused by B12 deficiency or by another malady. This issue may be resolved by requesting a test for one or more of the other available biomarkers of B12 deficiency, such as MMA or Hcy. A high level of MMA (adjusted for age and kidney function) by and large supports B12 deficiency, while a normal MMA makes it less likely [1,2].

Seeking an explanation for the increased value of total B12 may still be warranted, mainly to clarify possible analytical flaws, and to understand any relations between an increased level of total B12 and a pathophysiological condition of importance.

Analytical flaws are rare. As explained above, looking for an influence of heterophilic antibodies is warranted only in assays employing antibodies in the analytical process, which is uncommon for assays of B12. Currently, interference by IF autoantibodies is questionable, and further studies are needed to identify possible situations where it may be a concern.

Looking for macro-B12, employing nonspecific methods like PEG precipitation, is likely to be more misleading than helpful and cannot be recommended.

Obviously, treatment with B12 will result in an increased level of plasma B12. Excluding this cause, an increase in plasma total B12 is, as a rule of thumb, caused by an increased level of TC or HC, possibly in combination with high molecular components, such as autoantibodies toward these proteins. Thus, the optimal method for clarifying an unexpectedly high level of B12 is to measure the concentration of TC and HC. High concentrations of these proteins, notably observed in patients with levels of B12 well above the upper limit of the reference interval, have been related to liver and kidney diseases as well as malignancies, reviewed in refs. [22,26].

In conclusion, a possible B12 deficiency in patients with an unexpectedly normal/high level of B12 should be explored by analyzing MMA or Hcy, or both. A high level of B12 is seldom of clinical relevance, but if requested, it can be further explored by an immunological measurement of the B12 binding proteins. Looking for macro-B12 employing nonspecific methods, such as PEG precipitation, is problematic and should be avoided.

## Figures and Tables

**Figure 1 nutrients-16-00648-f001:**
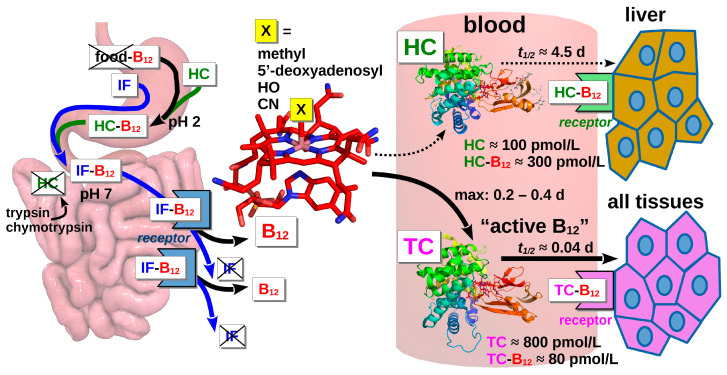
Scheme of B12 uptake. B12 present in food is liberated from its matrix and binds in the stomach to haptocorrin (HC), which is insensitive to low pH. Intrinsic factor (IF) is synthesized in the stomach but does not bind B12 at pH 2. In the small intestine, HC degrades, while B12 binds to IF after neutralization of the digest. The IF-B12 complex is internalized in the terminal ileum after binding to the cubam receptor, whereupon IF is degraded in lysosomes. Free B12 (red sketch in the middle) is exported to the blood and binds primarily to the fast-exchanging protein carrier transcobalamin (TC) due to its surplus binding capacity. The produced TC-B12 complex is afterward rapidly metabolized. A small quantity of B12 binds to the slowly metabolized carrier HC present at a lower concentration than TC. The slow clearance of the HC-B12 complex results in its considerable excess in blood plasma in comparison to the promptly removed TC-B12. Only the latter complex is regarded as “active B12”, readily available to all tissues. For further explanation, see the main text.

**Table 1 nutrients-16-00648-t001:** Measurement of total plasma B12 by automated methods, main features.

Description	Siemens: Advia, Centauer, Immulite	Roche: Cobas	Abbott: Architect	Beckman Coulter: Access
Type of B12 tracer (* B12)	Acridium labeled	Biotin coupled	Acridium labeled	Not labeled
Intrinsic factor (IF)	On paramagnetic particles, IF~▒	Ruthium labeled, * IF	On paramagnetic particles, IF~▒	Alkaline phosphatase-IF, * IF
Additional key reagent	None	Streptavidin, solid-phase	None	Solid goat anti-mouse, mouse anti-IF, does not bind to IF-B12
Assay design	Plasma B12 + * B12 + IF~▒	Plasma B12 + B12~▒ + * IF + streptavidin	Plasma B12 + IF~▒ washing + * B12	Plasma B12 + * IF + mouse antibody~▒ (recognizes apo-* IF)
Reference interval pmol/L	156–672	145–569	138–652	133–675

IF, intrinsic factor; ▒, solid phase; ~, connection to a solid phase; *, labeled compound used for quantification. The information presented has been retrieved from the kit inserts for the various methods.

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
