# Peer review of "Macro-B12 and Unexpectedly High Levels of Plasma B12: A Critical Review"

_nutrients, 2024, doi:10.3390/nu16050648_

Round 1
Reviewer 1 Report
Comments and Suggestions for Authors
The present manuscript discusses the challenges in interpreting plasma vitamin B12 levels, emphasizing the controversies surrounding unexpected normal/high levels. The concept of "macro-B12," defined as the fraction removable by polyethylene glycol (PEG) precipitation, is scrutinized. The presence of B12 in plasma, bound to transcobalamin and haptocorrin, is highlighted, and elevated concentrations in these proteins usually result in an increased total plasma B12. The automated competitive binding assays, often mistaken as immunoassays, are used for measurement, relying on intrinsic factor rather than antibodies. The manuscript suggests exploring unexpectedly high B12 levels through immunological measurements of haptocorrin and transcobalamin, cautioning against nonspecific methods like PEG precipitation. The need for evaluating high B12 levels of unknown origin is considered limited, as alternative tests, such as measuring methylmalonic acid, may offer better guidance for diagnosing B12 deficiency.
Comments:
In the article, it is stated that approximately 4/5 of circulating B12, along with all analogues, binds to HC. In healthy individuals, HC is present at a concentration of 240-680 pmol/L [17]. Additionally, about 1/5 of all plasma B12 is bound to TC. In healthy individuals, the total TC concentration ranges from 670-1270 pmol/L [20], but only around 10% of the protein is saturated with B12.
The concentration of TC is much higher than that of HC; this concept could be elaborated upon to explain why HC, despite its lower concentration, binds the majority (4/5) of vitamin B12.
At line 86, it is suggested to add that hepatocytes also have a receptor for TC in addition to the receptor for HC, and the functioning of this dual receptor presence is not well understood.
At line 90, it is mentioned that "TC is produced by endothelial cells and binds B12 with an affinity comparable to that of HC [18]." However, caution is advised, as the binding capacity of vitamin B12 with HC and TC differs. There is an exchange of vitamin B12 from HC to TC, and researchers are beginning to discuss the idea that the storage site for vitamin B12 in the human body is not only the liver but also the blood.
The bibliography appears somewhat outdated, lacking citations from 2023 and 2024. It is recommended to include some review articles that provide an updated overview of vitamin B12 deficiency and metabolic decompensation. Additionally, in line 27, where it is mentioned that vitamin B12 deficiency may lead to irreversible neurological damages, citing a reference from 2017, it is suggested to consider more recent review articles from 2024 if they offer updated insights into the diagnosis and management of vitamin B12 deficiency.
Comments on the Quality of English LanguageNo comments
Author Response
Reviewer 1
- Reviewer comment: The present manuscript discusses the challenges in interpreting plasma vitamin B12 levels, emphasizing the controversies surrounding unexpected normal/high levels. The concept of "macro-B12," defined as the fraction removable by polyethylene glycol (PEG) precipitation, is scrutinized. The presence of B12 in plasma, bound to transcobalamin and haptocorrin, is highlighted, and elevated concentrations in these proteins usually result in an increased total plasma B12. The automated competitive binding assays, often mistaken as immunoassays, are used for measurement, relying on intrinsic factor rather than antibodies. The manuscript suggests exploring unexpectedly high B12 levels through immunological measurements of haptocorrin and transcobalamin, cautioning against nonspecific methods like PEG precipitation. The need for evaluating high B12 levels of unknown origin is considered limited, as alternative tests, such as measuring methylmalonic acid, may offer better guidance for diagnosing B12 deficiency.
Answer: Thank you. - Reviewer comment: In the article, it is stated that approximately 4/5 of circulating B12, along with all analogues, binds to HC. In healthy individuals, HC is present at a concentration of 240-680 pmol/L [17]. Additionally, about 1/5 of all plasma B12 is bound to TC. In healthy individuals, the total TC concentration ranges from 670-1270 pmol/L [20], but only around 10% of the protein is saturated with B12. The concentration of TC is much higher than that of HC; this concept could be elaborated upon to explain why HC, despite its lower concentration, binds the majority (4/5) of vitamin B12.
Answer: We have modified the related paragraph to address this issue: “IF recognizes B12 and facilitates its intestinal uptake via interaction with an IF-B12-specific receptor (Figure 1). At the same time, IF has a low affinity for analogues [18], which are expected to pass to thick intestine without any significant adsorption [18]. B12 from dietary sources eventually enters blood, topping at approximately 7 h after ingestion, see Figure 1 and ref. [18] for a review. The incoming vitamin primarily binds to TC due to its high binding capacity (600-1200 pmol/L [18,20]), maintained by a steady production of TC in endothelial cells. TC has an affinity for B12, which is comparable to that of HC, but the binding of analogues to TC is relatively weak. The produced TC-B12 complex (often referred to as holoTC or “active B12”) is promptly absorbed by human tissues (t½ ≈ 1 h [21]), and particularly by the liver. A fast clearance of TC-B12 stipulates its low steady state concentration (≈ 60 pM), which corresponds under normal conditions to 10 – 20 % of total plasma B12. A low level of holoTC contrasts a high concentration of the “inert” HC-B12 complex, and this disproportion occasionally creates confusion during assessment of B12 status by total plasma B12. Therefore, a separate test on holoTC in blood is currently implemented in many medical laboratories [1–4].”
- Reviewer comment: At line 86, it is suggested to add that hepatocytes also have a receptor for TC in addition to the receptor for HC, and the functioning of this dual receptor presence is not well understood.
Answer: The quantity of B12 delivered to the liver by HC is small, and the major part is delivered by TC through TC specific receptors. We have clarified this in our answer to Comment 2.
- Reviewer comment: At line 90, it is mentioned that "TC is produced by endothelial cells and binds B12 with an affinity comparable to that of HC [18]." However, caution is advised, as the binding capacity of vitamin B12 with HC and TC differs. There is an exchange of vitamin B12 from HC to TC, and researchers are beginning to discuss the idea that the storage site for vitamin B12 in the human body is not only the liver but also the blood.
Answer: We have provided additional explanations in our answer to Comment 2. We do not think that blood can serve as a storage of B12, because the content of B12 in blood plasma (approximately 1.5 µg) is by far lower than the content in the tissues (3000 – 5000 µg), where approximately one half is stored in the liver, see e.g. ref. 34. The blood is rather a vehicle of delivery and exchange of B12 between different tissues than its storage according to this material balance.
- Reviewer comment: The bibliography appears somewhat outdated, lacking citations from 2023 and 2024. It is recommended to include some review articles that provide an updated overview of vitamin B12 deficiency and metabolic decompensation. Additionally, in line 27, where it is mentioned that vitamin B12 deficiency may lead to irreversible neurological damages, citing a reference from 2017, it is suggested to consider more recent review articles from 2024 if they offer updated insights into the diagnosis and management of vitamin B12 deficiency.
Answer: The bibliography reflects a revival of older data that correctly explore a high B12 to emphasize the problems occurring when using newer and nonspecific methodologies. We agree that a new general review may be warranted and have exchanged ref 2 with a recently published review. In such a way, we have also decreased the rate of self-citation, requested by the Editor. A few other modifications in the references have been done.
Reviewer 2 Report
Comments and Suggestions for Authors
The reviewed manuscript discusses the determination of vitamin B12 level in plasma and the interpretation of results. In my opinion, the paper is well-written and significant from a medical standpoint regarding the discussed topic's relevance. The authors have provided a wide range of valuable information concerning factors that may affect the level of vitamin B12 in serum. The work largely focuses on proteins binding to vitamin B12 and "macro-B12" complexes. Before publication, I suggest that the authors address more precisely the issue of the different forms of B12: methyl-, hydroxo-, cyano-, and adenosylcobalamin. Do these individual forms differ in their affinity to binding proteins, and could this have implications for result analysis and interpretation? This is an interesting issue due to the availability of various drugs and supplements used in the prevention and treatment of B12 deficiencies.
Author Response
Reviewer 2
- Reviewer comment: The reviewed manuscript discusses the determination of vitamin B12 level in plasma and the interpretation of results. In my opinion, the paper is well-written and significant from a medical standpoint regarding the discussed topic's relevance. The authors have provided a wide range of valuable information concerning factors that may affect the level of vitamin B12 in serum. The work largely focuses on proteins binding to vitamin B12 and "macro-B12" complexes.
Answer: Thank you. - Reviewer comment: Before publication, I suggest that the authors address more precisely the issue of the different forms of B12: methyl-, hydroxo-, cyano-, and adenosylcobalamin. Do these individual forms differ in their affinity to binding proteins, and could this have implications for result analysis and interpretation? This is an interesting issue due to the availability of various drugs and supplements used in the prevention and treatment of B12 deficiencies.
Answer: A relevant comment has been included: “… and bind with comparable affinities to the B12 binding proteins [14,15].” A more detailed comparison of CNCbl, AdoCbl and HOCbl is presented in doi: 10.1074/jbc.M111399200 and 10.1021/bi062063l. Unfortunately, we cannot refer to these works because a very strict limit of self-citation in Nutrients precludes us from adding more references to our own publications.
In addition, minor language revision has been made.